# Analysis of Vertical Distribution Changes and Influencing Factors of Tropospheric Ozone in China from 2005 to 2020 Based on Multi-Source Data

**DOI:** 10.3390/ijerph191912653

**Published:** 2022-10-03

**Authors:** Yong Zhang, Yang Zhang, Zhihong Liu, Sijia Bi, Yuni Zheng

**Affiliations:** 1College of Resources and Environment, Chengdu University of Information Technology, Chengdu 610225, China; 2Meteorological Service Center of Xinjiang Uygur Autonomous Region, Urumqi 830002, China; 3Key Laboratory of High Power Laser and Physics, Shanghai Institute of Optics and Fine Mechanics, Chinese Academy of Sciences, Shanghai 201800, China

**Keywords:** tropospheric ozone, OMI, remote sensing vertical monitoring, spatiotemporal change, cause analysis

## Abstract

The vertical distribution of the tropospheric ozone column concentration (OCC) in China from 2005 to 2020 was analysed based on the ozone profile product of the ozone monitoring instrument (OMI). The annual average OCC in the lower troposphere (OCC_LT_) showed an increasing trend, with an average annual increase of 0.143 DU. The OCC in the middle troposphere showed a downward trend, with an average annual decrease of 0.091 DU. There was a significant negative correlation between the ozone changes in the two layers. The monthly average results show that the peak values of OCC_LT_ occur in May or June, the middle troposphere is significantly influenced by topographic conditions, and the upper troposphere is mainly affected by latitude. Analysis based on multi-source data shows that the reduction in nitrogen oxides (NO_x_) and the increase in volatile organic compounds (VOCs) weakened the titration of ozone generation, resulting in the increase in OCC_LT_. The increase in vegetation is closely related to the increase in OCC_LT_, with a correlation coefficient of up to 0.875. The near-surface temperature increased significantly, which strengthened the photochemical reaction of ozone. In addition, the increase in boundary layer height also plays a positive role in the increase in OCC_LT_.

## 1. Introduction

Ozone is a trace gas in the Earth’s atmosphere, accounting for only 0.0012% of the atmospheric composition [1]. Ozone is a strong absorber of ultraviolet solar radiation and most of the ozone concentrated in the stratosphere protects the Earth’s flora and fauna from UV damage [2,3]. However, if the concentration of ozone in the lower troposphere is too high, it can have serious effects on plants and animals. For example, high ozone concentrations can also inhibit chlorophyll synthesis, reducing the intensity of photosynthesis and reducing crop yields, among other effects [4,5]. Prolonged human exposure to ozone exceedances can also cause damage to the heart and other organs, cause acute respiratory infections, or lead to a reduction in human life expectancy and increased mortality [6]. Some projections suggest that if convective ozone pollution continues to increase, global ozone pollution could cause a total loss of USD 12–21 billion per year in cash crop yields and pose an increasing threat to global food security [7]. Consequently, tropospheric ozone has become a hot topic of research in various fields.

Photochemical smog was first discovered in the late 1970s in the Xigu petrochemical area of Lanzhou [8], Photochemical smog was also detected in Beijing in 1986, so in the 1980s, the China Meteorological Administration (CMA) established atmospheric background benchmark monitoring stations in the areas of Beijing-Tianjin-Hebei (BTH), Yangtze River Delta (YRD), and the Northeast Plain to continuously monitor atmospheric ozone concentrations over a long period of time. Since then, scholars have carried out integrated studies on atmospheric physics and chemistry using stations to monitor ozone pollution in cities such as Beijing, Guangzhou, Hong Kong, and Lin’an [9,10]. Xu et al. found that the maximum monthly average ozone concentration at background sites increased at a rate of 1.8 ppbv per year from 1991 to 2006 in Eastern China, and the increase in nitrogen oxides (NO_x_) led to an increase in ozone concentration [11]. Tang et al. found that the maximum 1 h daily concentration of ozone in urban areas increased by 1.3% per year from 2001 to 2006 in Beijing, while the maximum 8 h daily concentration of ozone in suburban areas increased by 1.1 ppbv per year from 2003 to 2015, and it was also found that reductions in NO_x_ emissions and increases in non-methane hydrocarbons emissions lead to increased ozone concentrations [12]. Chao He et al. investigated the meteorological causes of ozone pollution in China using data from environmental monitoring sites, and showed that near-ground ozone concentrations in China increased year-by-year and that temperature was the main meteorological driver of warm season changes in China, with its influence on concentrations in northern, north-western, and north-eastern China being significantly higher than in other regions [13]. NO_x_ has a significant titration effect on ozone, and ozone concentration decreases exponentially with increasing NO_x_ concentration. High temperature and low humidity favour the formation of ozone. Similar to NO_x_, ozone concentration increases and decreases exponentially with increasing temperature and relative humidity [14].

With the development of satellite technology, the technique of monitoring the spatial and temporal distribution of ozone in the atmosphere by satellite remote sensing has gradually matured, and the characteristics of remote sensing technology, such as high efficiency, long duration, and holistic surface monitoring, have well compensated for the lack of spatial and temporal range of station monitoring, allowing it to be widely used in atmospheric environmental monitoring. Noreen et al. used ozone monitoring instrument (OMI) and microwave limb sounder (MLS) data to study the spatial distribution and temporal evolution of the total organic carbon (TOC) over Pakistan from 2004 to 2014; the results show that TOC increased by 3.2 ± 1.1 DU over Pakistan. Ozone concentration exhibits a significant positive correlation with the seasonality of UV-B flux, NO_x_, and volatile organic compounds (VOCs), and VOCs emitted by organisms with temperature dependence [15]. The analysis of Chen et al. using tropospheric ozone from the OMI/MLS shows a consistent increase in tropospheric ozone from 2005 to 2017, strongly related to meteorological factors such as precipitation, surface temperature, planetary boundary layer height (PBLH), and horizontal winds [16]. Li et al. used OMI to analyse the ozone change in BTH and its surrounding areas during 2005–2018, and found that the surface ozone concentration showed an increasing trend, with an average growth rate of 3.4 μg m^−3^y^−1^, and with a greater increase in the second half of each year than the first. This is mainly due to the stronger photochemical reactions caused by a sustained increase in HCHO and a rapid decrease in NO, resulting in a weakening of the titration effect [17]. Hung et al. used Nimbus-4 data in combination with sounding data to analyse the relationship between the spatial distribution of ozone and PBLH in the Canadian region, and showed that PBLH affects tropospheric ozone distribution [18]. Zhao et al. studied ozone pollution events in industrial cities in Xuzhou, Nanjing, Shanghai, and Hangzhou, where increases in ozone concentrations were often accompanied by higher temperatures, while the response to humidity was not significant [19]. Du et al. used OMI to analyse the characteristics of total ozone column concentration (OCC) in China, and found that the spatial distribution characteristics of total ozone in China are high in the north and low in the south, high in the east and low in the west, and low in summer and autumn and high in winter and spring. In the Qinghai-Tibet Plateau (QTP), there is a trough of ozone in summer and autumn [20]. Zhu et al. used OMI data to extract and analyse the tropospheric OCC in China from 2005 to 2019, and the results show that the ozone concentration in China decreased from the northeast to the southeast, and decreased sequentially in spring, winter, summer, and autumn. The tropospheric OCC in southwest China was significantly positively correlated with temperature, wind field, and vegetation coverage; NO_x_ and VOCs were significantly positively correlated; and precipitation was significantly negatively correlated. Temperature, wind field, NO_x_, and VOCs emissions are key factors [21].

As shown above, more studies on tropospheric ozone in China have been conducted based on ground-based monitoring stations and satellite remote sensing, and some scholars have also analysed the relationship between changes in tropospheric ozone content and meteorological elements and ozone precursors, gaining a preliminary understanding of the current status of tropospheric ozone pollution in China. However, ground-based monitoring stations are limited by their spatial coverage, making it difficult to monitor ozone with continuous spatial coverage, which poses certain limitations to the in-depth understanding of the spatial and temporal distribution of regional ozone. The vertical variation of ozone in the troposphere is not known, and there is no way to know whether the upper troposphere affects the ozone concentration in the near-Earth layer. Most of the existing remote-sensing-based studies of tropospheric ozone investigated the spatial and temporal variability of ozone in the troposphere as a whole, but no studies of the vertical stratification of tropospheric ozone have been carried out. There is no way of knowing the vertical variation in ozone within the troposphere, let alone whether the upper troposphere affects ozone concentrations in the near-Earth layer. To fill the gaps in the existing literature, this study used OMI ozone profile products to analyse the spatiotemporal variation of the tropospheric OCC in China, and multi-source data were used to analyse the causes of the OCC changes. We hope that this study will deepen the current understanding of ozone problems in China.

## 2. Data and Methods

### 2.1. Overview of the Study Area

In this paper, eight major urban agglomerations in China were selected as the study area, including BTH, YRD, QTP, Urumqi-Changji-Shihezi (UCS), Pearl River Delta (PRD), Sichuan Basin (SCB), Centre of China (COC), and Mid-Southern Liaoning (MSL). The specific division and the corresponding spatial ranges of these regions are shown in Table 1 and Figure 1. For ease of presentation, the abbreviations are used to indicate each region in the following text.

### 2.2. Data Introduction

The ozone profile data come from OMI and can be obtained from the Earthdata website (https://search.earthdata.nasa.gov/, accessed on 2 October 2022). The sensor is mounted on Arua, NASA’s third new-generation Earth observation system satellite. It is mainly used to observe the global atmospheric composition and belongs to a Hyperspectral meter. The OMI ozone profile data (OMO3PR) used in this study, which divide the atmosphere into 18 layers according to atmospheric pressure, represent the observations of OCC from the near-surface to 0.3 hPa at different altitudes, given in Dobson units (DU) [22]. A 10-micron-thick ozone layer in standard atmospheric conditions is represented by 1 DU. Research shows that OMI’s measurement error in the global troposphere is between 2.4 and 3.1 DU [23]. Ozone retrieval accuracy ranges from 1% in the middle stratosphere to 10% in the lower stratosphere and lower troposphere. The error is between 1 and 6% in the middle of the stratosphere and is 6–35% in the troposphere, mainly due to smoothing errors. In China, the results regarding OMI tropospheric ozone have strong consistency with the results in the north and south regions of the near-surface and surface stations, which indicates that the use of OMI data can well represent the characteristics of tropospheric OCC [24]. This paper mainly uses the tropospheric data of the 15th to 18th layers from 2005 to 2020. The corresponding height information of each layer is shown in Table 2, which, respectively, represents the OCC of the four altitude layers from the near-surface (0 km) to 12.4 km.

In this study, multi-source data such as ozone precursor satellite remote sensing products, vegetation cover satellite remote sensing products, and meteorological reanalysis data were used to analyse the causes of changes in tropospheric ozone. The time period of the multi-source data is from 2005 to 2020, which is consistent with the time of the OMI ozone profile product. The remote sensing product data of ozone precursors are NO_2_ and HCHO. The tropospheric NO_2_ data are the OMI tropospheric column concentration product data provided by Goddard Earth Sciences Data and Information Services Center (GES DISC) with a resolution of 0.25° × 0.25°. The tropospheric HCHO data with a resolution of 0.1° × 0.1° are the data released by ESA’s TEMIS project. It adopts DOAS technology combined with radiation transfer for calculation, and uses the results of the IMAGES chemical transfer model as prior information. Its data quality is significantly improved compared with the accuracy of OMI’s HCHO products, between 2010 and 2016. The meteorological reanalysis data comes from the ERA-5 global reanalysis data provided by the European Centre for Medium-Range Weather Forecasts. This paper uses the air temperature and PBLH data set at the heights of 1000, 950, 900, 850, 800, and 750 hPa. In order to ensure the scientific nature of the analysis results, the ERA-5 data at a time similar to the OMO3PR data set was used, that is, the data at 6:00 UTC time, with a spatial resolution of 0.25° × 0.25°. This study shows that, compared with the ground observation data, the ERA-5 results are more consistent in mainland China [25]. In addition, this paper uses the MYD13Q1 vegetation index product of MODIS to explore the effect of vegetation growth on ozone generation. The data are derived from GES DISC with a spatial resolution of 1 km × 1 km.

### 2.3. Research Methodology

#### 2.3.1. Data Pre-Processing

For the ozone profile data set, the daily results of OCC in different height layers were reprojected to the latitude–longitude grids with a spatial resolution of 0.3° × 0.3°. After that, the daily data were averaged to form monthly, quarterly, and annual data sets for statistical analysis, and the 0 values and invalid values were not included during the calculation of the average values (same below). Because the NO_2_, HCHO, and ERA-5 data are results in the form of latitude–longitude grids, the mean value statistics can be performed directly. For the MODIS vegetation index product, the “VI Quality” data set in the product is mainly used to remove the outliers due to the influence of snow and clouds, and the results after removing the outliers are converted from sinusoidal projection to the form of latitude–longitude grids for mean statistics. To ensure a consistent data range, the above multi-source data were clipped using the China administrative region vector, and the results over mainland China were retained. All the above processes were implemented through Interactive Data Language (IDL).

#### 2.3.2. Statistical Analysis

In this study, the mean year-on-year change (*MYC*) was used to characterise the changes in OCC in different height layers of eight major urban groups to clarify the absolute changes in their OCCs over a 16-year period. It is calculated as follows:(1)$$MYC=\frac{\sum_{i=2}^{n}\left(S_{i}-S_{i-1}\right)}{n}\;$$
where $S_{i}$ denotes the data in year i and n denotes the duration of the data.

This study mainly uses the slope trend analysis to characterise the spatial and temporal trends of various types of data. It is calculated as follows:(2)$$Slope=\frac{n\sum_{i=1}^{n}\left(i\times S_{i}\right)-\sum_{i=1}^{n}i\times \sum_{i=1}^{n}S_{i}}{n\sum_{i=1}^{n}i^{2}-{\left(\sum_{i=1}^{n}i\right)}^{2}}$$
where Slope represents the slope of the pixel-by-pixel regression equation, $S_{i}$ denotes the i-th year data, and n denotes the duration of the data. When Slope > 0, it means that the data $S_{i}$ has an increasing trend; when Slope < 0, it means that the data $S_{i}$ has a decreasing trend; and when Slope = 0, it means that the data $S_{i}$ has no significant change.

In this study, Pearson’s correlation coefficient was used to characterise the correlation between the OCC and the other environmental factors, which was calculated as follows:(3)$$R_{i,j}=\frac{\sum_{i=1}^{n}\left(x_{i}-\bar{x}\right)\left(y_{i}-\bar{y}\right)}{\sqrt{\sum_{i=1}^{n}\left(x_{i}-\bar{x}\right)}\sqrt{\sum_{i=1}^{n}\left(y_{i}-\bar{y}\right)}}$$
where n is the total number of samples, $x_{i}$ and $y_{i}$ denote the ith element and OOC samples, respectively, and $R_{i,j}$ characterises the Pearson correlation coefficient between the x and y factors.

## 3. Results and Analysis

### 3.1. Characteristics of the Vertical Spatial Distribution of Tropospheric Ozone

The vertical tropospheric ozone profiles of the eight urban groups are shown in Figure 2, where the red line represents the mean value for the whole region of China. From an overall perspective, the variation pattern of tropospheric OCC in the vertical stratification column is rising first and then decreasing, with the OCC at 0–3 km height being 7.6 ± 0.86 DU; up to 3–5.8 km, the OCC increases slightly and the concentration is 9.5 ± 1.10 DU; at 5.8–9.6 km height, the OCC is 10.8 ± 0.49 DU; and at 9.6–12.4 km, the concentration decreases slightly by 8.7 ± 1.02 DU.

With the exception of QTP, the profile trends of the other seven study areas are more consistent with the national ozone profile changes, showing an overall trend of being higher in the east and lower in the west, with the lowest overall OCC in QTP and the fastest increase in concentration at 5.8–9.6 km; in the remaining seven study areas, the overall OCCs of BTH and MLS were the highest and increased rapidly from 3 to 5.8 km, while those of COC, YRD, and PRD changed less, increasing only slightly from 3 to 5.8 km and decreasing slightly from 5.8 to 9.6 km; SCB changed less from 0 to 5.8 km and increased slightly from 5.8 to 12.4 km with altitude, and UCS increased from 0 to 9.6 km and decreased slightly from 9.6 to 12.4 km. The concentration of UCS at 0.6–9.6 km increased with increasing altitude and decreased slightly at 9.6–12.4 km.

#### 3.1.1. Characteristics of the Horizontal Distribution of Tropospheric Ozone at Different Heights

Figure 3a shows the distribution of OCC levels in mainland China from 0 to 3 km, and clearly shows that the distribution of ozone in this layer in China is more obviously distributed according to the topographic trend, and the national average OCC is 7.6 DU. The high value ozone zone is distributed in BTH, YRD, COC, and SCB, with an OCC around 12.5 DU; the second high-value zone is distributed in PRD, MLS, and UCS, with an OCC around 5–7 DU; and the low-value zone is mainly in QTP, with a relatively small OCC, being only around 0–1 DU. It is noted that the average elevation in the QTP region is above 4 km, making the 0–3 km OCC results lower in this region. As can be seen from Figure 3b, the trend in the topographic distribution of ozone remains more pronounced at altitudes of 3–5.8 km, with national OCC at this layer increasing by approximately 3 DU compared with 0–3 km, with an average concentration of 10.3 DU. The high-value area also shows a clear clustering distribution, with the relative range of the high value area decreasing, mainly concentrated in BTH, MLS, UCS, and the northern part of YRD, where the concentration is around 13 DU; the secondary high-value area is mainly located in SCB, PRD, eastern COC, and the southern part of YRD, with an OCC around 11 DU; and QTP remains the lowest value area, with an OCC of only 2–3 DU. As shown in Figure 3c, although the OCC at 5.8–9.6 km is still influenced by topography (e.g., the QTP region), it also shows a latitude effect, with the OCC at higher latitudes being significantly higher than at lower latitudes. For example, the OCC in the MSL area can exceed 12 DU, while that in the PRD area is only around 9 DU. The OCC in the QTP region remains the lowest, but has grown to the range of 8–9 DU. Finally, the results of Figure 3d show that the spatial distribution of OCC at 9.6–12.4 km is hardly affected by topography, and latitude is the main factor affecting the OCC at this layer. The high values are found between 45° N and 54° N, with concentrations around 12–15 DU; while the overall OCC in southern China is less than 7 DU, the largest difference between the north and south can reach 10 DU.

#### 3.1.2. Seasonal Distribution Characteristics of Tropospheric Ozone at Different Heights

The seasonal distribution of tropospheric ozone profiles in mainland China from 2005 to 2020 is shown in Figure 4. The seasons are defined as spring from March to May, summer from June to August, autumn from September to November, and winter from December to February. From the figure, it can be seen that there are large seasonal differences in the OCC. In spring, the OCC increases with altitude, from 8.22 ± 1.05 DU at 0–3 km to 11.84 ± 1.30 DU at 9.6–12.4 km. In summer, the OCC shows a trend of increasing first and then decreasing with height, with the maximum OCC value of 11.25 ± 0.95 DU appearing at the height of 5.8–9.6 km. The OCC in autumn shows a similar trend to summer, but the maximum value of OCC occurs at 3–5.8 km, with a value of 9.25 ± 1.19 DU. In winter, the maximum value of OCC also occurs at 3–5.8 km, with a value of 10.50 ± 0.45 DU, and there was no significant difference in OCC between 5.8–9.6 km and 9.6–12.4 km. At 0–3 km, the value of OCC falls in descending order as summer > spring > winter > autumn, with a maximum value of 8.80 ± 1.29 DU in summer. For the remaining altitudes, the highest OCC always occurs in spring. However, it should be noted that the OCC only has a small standard deviation range in winter and, in other seasons, the OCC at different altitudes may change significantly.

### 3.2. Time-Varying Characteristics of the Vertical Profile of Tropospheric Ozone

#### 3.2.1. Characteristics of Interannual Variability

As can be seen from Figure 5, the tropospheric OCC in mainland China has generally shown a continuous increase over the past 15 years, from 32.0 DU in 2005 to 34.52 DU in 2020, with a growth rate of 7.9%. The segmental trends of tropospheric OCC can be summarised as follows: the period 2005–2008 showed little change; the period 2009–2018 showed a continuous increase, with tropospheric ozone OCC rising from 32.26 to 35.62 DU, an increase of 3.36 DU or 10.4%; and the period 2019–2020 showed a decreasing trend, with a decrease of 1.1 DU or 3%. The two years with the largest increases in tropospheric OCC were 2009 and 2018, with increases of 2.88 and 2.47%, respectively, compared with the previous year.

Figure 6 shows the annual mean changes in OCCs at different altitudes for the eight major urban agglomerations from 2005 to 2020, and Table 3 shows the corresponding mean year-on-year changes (MYC) in OCCs. At the 0–3 km altitude level, the OCC in the eight major urban agglomerations show an increasing trend, with a regional MYC of 0.143 DU, among which BTH, MSL, COC, and YRD show larger increases, with MYCs greater than 0.15 DU. In contrast to the 0–3 km layer, the OCCs in the eight major urban clusters at 3–5.8 km all showed a decreasing trend, with a regional MYC of −0.09, the decrease being smaller than the increase at 0–3 km, and the decrease being proportional to the increase at 0–3 km. Here, it is hypothesised that 3–5.8 km is one of the sources of the increase in 0–3 km OCCs, a conjecture that will be tested later. For the 5.8–9.6 km altitude layer, there is no clear pattern of variation in OCCs relating to geography, and the overall variation is small, lying mainly in the range of −0.05 to +0.03 DU. For the 9.6–12.4 km altitude layer, the OCCs, except for SCB, show a decreasing trend, and the variation is still small. It is assumed that the variation in OCCs in the 5.8–9.6 km and 9.6–12.4 km altitude layers is mainly due to cyclical fluctuations caused by climatic factors. For example, changes in atmospheric circulation can cause material exchange between the bottom of the stratosphere and the top of the troposphere, resulting in small disturbances in the OCC [26].

#### 3.2.2. Characteristics of Monthly Variation

Figure 7 shows the monthly average trend of OCCs in different altitudes of the troposphere in different regions of China from 2005 to 2020. From this, it can be seen that the trend of 0–3 km OCCs in the seven regions, except QTP, is relatively consistent, showing a single-peak distribution in general, i.e., OCCs increase month-by-month in the first half of the year, reaching a yearly peak in May or June (SCB and PRD reach their peak in May, the rest of the regions in June), and then begin to gradually decline, reaching a yearly minimum in December. For QTP, the overall monthly trend in OCC has a continuous “M” pattern, but with small month-on-month changes. As can be seen in Figure 7b, the monthly variation in OCCs at 3–5.8 km varies greatly among regions: PRD and YRD have similar trends, both of which are in river delta topography; BTH, MSL, and COC have similar trends, all of which are in plain topography; and the remaining three regions have no similar characteristics, with SCB in hilly basin topography, QTP in highland topography, and UCS in desert basin topography. Therefore, the monthly variation in OCC at 3–5.8 km height is mainly dominated by the topographic conditions of the subsurface. As can be seen in Figure 7c, the OCCs at 5.8–9.6 km show an overall single-peaked distribution, with a gradual increase from January to April, with the peak in each region occurring between April and June, followed by an overall decreasing trend. As shown in Figure 7d, the trend of OCCs between 9.6 and 12.8 km is consistent, showing a sinusoidal distribution with a gradual increase from January to March, with peaks in each region occurring between March and May, decreasing from May to September and increasing from October to December.

## 4. Discussion

In Section 3.2.1, it is shown that the OCC in the lower troposphere (OCC_LT_) in the eight major urban agglomerations shows an increasing trend year-by-year. Figure 8 further shows the trend of the annual average OCC_LT_ for the whole land area of China from 2005 to 2020, which shows that the overall OCC_LT_ for the whole region of China is still exhibiting an increasing trend, with an overall concentration increase of 2%, and an average annual increase of 0.13 DU. The lower tropospheric ozone has a significant impact on human production and life, and the growth of plants and animals, so it is important to investigate the causes of the increase in order to combat ozone pollution. Studies have shown that ozone formation in the atmosphere is closely related to the ratio of VOCs to NO_x_. While VOCs have an obvious linear relationship with HCHO, hydrocarbons (RH) are oxidised by organic peroxyl radicals (RO_2_) and OH generated in the first stage of oxidation, which react with NO_x_ to produce HCHO or higher carbonyl groups. The subsequent reaction eventually produces HCHO [27,28], which can indirectly reflect the accumulation of VOCs through the properties of HCHO [29]. At the same time, due to China’s rapid development resulting in consistently high NO_x_ concentrations in China, the increase in VOCs has been the main driver of soaring ozone pollution across China due to the positive impact of rising HCHO concentrations on OCCs in the context of the Chinese government’s vigorous NO_x_ abatement policies and tight restrictions on NO_x_ emissions [30,31]. A large proportion of biogenic VOCs (BVOCs), which are ozone-generating precursors, originate from vegetation and other biogenic emissions [4,32]. Whereas BVOCs are more reactive in photochemical reactions, ozone is more sensitive to changes in BVOCs emissions and BVOCs contribute relatively more to ozone formation, making them an important precursor to tropospheric ozone production [33]. The analysis of vegetation changes in Chinese regions over many years based on satellite remote sensing data can help us to analyse the causes of low-level ozone growth; temperature is the most important meteorological factor for ground-level ozone concentrations across China, and ozone production is positively correlated with sensitivity to increased temperature, with temperature being one of the main causes of ozone production, and areas where near-surface ozone increases significantly often being accompanied by severe near-surface warming [34,35]. In terms of the direction of influence, temperature has a persistent positive effect on ground-level ozone concentrations in most cities [36]. At the same time, warming will not only increase the reaction rate, but also increase the natural emission of VOCs from natural sources, which contributes to ozone production [37]. In addition, it has been pointed out that under certain weather systems, the rise in PBLH will lead to the transport of ozone from the upper atmosphere to the near-surface layer [38], which will facilitate the vertical exchange of ozone-producing precursors within the boundary layer [39], and eventually lead to an increase in ozone concentration. Furthermore, studies have shown that the positive correlation between the PBLH and the concentration of near-surface ozone is even greater than that of ozone precursors, especially when high ozone concentrations occur [40,41]. The above studies show that the effect of PBLH on ozone concentration is opposite to that of particulate matter [42]. In summary, in this paper, four factors, namely ozone-producing precursors (HCHO, NO_2_), normalised difference vegetation index (NDVI), air temperature, and PBLH, were selected to carry out a causal analysis of the increase in the OCC_LT_.

### 4.1. Ozone Transport in the 3–5.8 km Altitude Layer

Figure 9 shows the spatial distribution of the OCC slope in the 0–3 km and 3–5.8 km altitude layers in China from 2005 to 2020. Note that due to the influence of altitude, only a few areas of the QTP have valid pixels, so the slope of most areas is 0. Except for the QTP, the OCC_LT_ mainly shows an increase, while that at the 3–5.8 km layer mainly shows a decrease. The spatial distribution of the two shows an obvious inverse distribution, so the 3–5.8 km layer most likely contributes to the increase in the OCC. To determine this possibility, an analysis of whether the 3–5.8 km altitude layer transmits ozone upwards is required. In this paper, the slope of all altitude layers of the OMI ozone profile product for 2005–2020 was calculated. The slope of the OCC at each altitude layer and 3–5.8 km was correlated, and the corresponding correlation coefficient results are shown in Table 4. Because the number of effective pixels in the 0–3 km altitude layer is smaller due to the altitude of the QTP, the total sample size of the 0–3 km altitude layer is smaller than that of the other altitude layers. The table shows that although there is a high correlation between several altitude layers and the 3–5.8 km OCC slope, a positive correlation coefficient only represents a homogeneous change in slope, i.e., the OCC is decreasing in all these altitude layers, and if the 3–5.8 km altitude layer has a direct influence on these altitude layers, the correlation coefficient should be negative. There are four altitudes with negative correlation coefficients: 24–26.7 km, 21–24 km, 18.8–21 km, and 0–3 km. The correlation coefficient for 0–3 km is −0.891, indicating that there is an obvious negative correlation between the 3–5.8 km and 0–3 km OCCs. The correlation coefficients for the rest of the height layers are all greater than −0.3, and the negative correlation is relatively weak. The above analysis shows that the 3–5.8 km altitude layer contributes significantly to the increase in OCC_LT_, i.e., there is a transfer of ozone from the 3–5.8 km altitude layer to the 0–3 km altitude layer.

However, as can be seen from the mean year-on-year changes in OCCs at different altitudes in the troposphere for each region in Table 4, the decrease in OCCs at altitudes of 3–5.8 km is not sufficient to offset the increase at altitudes of 0–3 km, so there are still other external factors influencing OCC_LT_. Next, analysis will be carried out in relation to ozone precursors, vegetation cover, meteorological conditions, and other factors.

### 4.2. Precursor Effects

The annual trend of HCHO and NO_2_ tropospheric variation over China from 2005 to 2020 is shown in Figure 10. As seen from Figure 10a, the growth trend of HCHO during 2005–2016 is obvious. The average slope is 0.072, and the maximum value reaches 0.72, with 82.7% of the pixels showing positive growth. The high HCHO slope area is mainly in eastern China, including BTH, MSL, YRD, COC, and SCB. In the south-central region of QTP and Yunnan Province, although the increase in HCHO concentration is obvious, it is not conducive to ozone production because of the lower temperature and higher level of water vapour. From Figure 10b, it can be seen that the trend of NO_2_ changes in the east and west of China during 2005–2020 differed significantly; the background value of NO_2_ in western China increased slowly, with its slope being between 0 and 0.1 overall, and between 0.1 and 0.2 in a few areas; southern and central-eastern China showed a significant decreasing trend, with the PRD, YRD, COC, and BTH areas decreasing most significantly, exhibiting slope indices greater than −0.6, and up to −0.93. SCB, MLS, and UCS also have a more obvious reduction, which is related to the country’s recent introduction of relevant energy-saving and emission-reduction measures. In a comprehensive analysis, comparing the spatial distribution trends of HCHO slope growth areas with Figure 6a, there is an obvious agreement between the two, especially in MLS, BTH, YRD, COC, and SCB, with a trend correlation up to r = 0.8, which indicates that the continuous growth of HCHO is an important driver of the rising OCC_LT_ in China. At the same time, the rapid decline in NO_2_ is also in high agreement with the spatial distribution trend of 0–3 km ozone. The areas where NO_2_ declines significantly are also the areas where the distribution of high value ozone areas is concentrated, most significantly in PRD, YRD, COC, and BTH, while the areas where NO_2_ changes insignificantly or increases also have relatively low ozone concentrations. Previous studies have shown that when atmospheric NO_x_ is at high concentrations, the reduction in NO_x_ emissions will reduce the NO_x_ titration and the reduction of ozone by NO_x_ will also increase the oxidation of VOCs to produce ozone and OH, resulting in higher ozone concentrations [43,44]. Figure 10a,b shows that the increasing HCHO level and decreasing NO_2_ level in China over the years have disrupted the mutual balance of ozone production and depletion by atmospheric VOCs and NO_x_, and the “titration” of ozone by NO has weakened, resulting in a shift from VOC-limited to VOC-driven ozone generation [45,46,47]. This has resulted in more conversion of VOCs to ozone, leading to a continued increase in tropospheric near-surface ozone concentrations, suggesting that NO_x_ and VOCs should be controlled in tandem, rather than one or the other.

### 4.3. Vegetation Growth Effects

Monthly mean values of the NDVI for 2004–2020 were obtained based on the MYD13Q1 product and compared with the OCC_LT_, the results of which are shown in Figure 11. Over the years, NDVI has been on a fluctuating upward trend in most regions of China, while NDVI and OCC_LT_ show a cyclical variation with obvious consistency in all regions except QTP. During March to July of each year, vegetation grows rapidly with increasing temperature, and also releases a large amount of BVOCs, including monoterpenes and isoprenes [48], which also provide good conditions for photochemical reactions to generate ozone [49]. From July to August, when vegetation is flourishing and growing more slowly, the release of BVOCs decreases and ozone concentrations begin to decrease; from September to February, as vegetation gradually dies, NDVI drops rapidly to lower values, and along with the rapid drop in temperature, ozone concentrations also begin to decrease rapidly to their lowest value of the year [50,51]. At the same time, although OCC_LT_ increases with NDVI, it also shows a significant lag, with peak NDVI lagging behind peak OCC_LT_ by approximately one month in most areas, especially in BTH, MLS, COC, SCB, and UCS. If the peak NDVI is used as a marker of plant maturity, the stage of NDVI increase can correspond to the growing phase of plants, which in turn indicates that plants drive OCC_LT_ in these areas more strongly during the growing phase than during maturity. In the PRD region there is a 2-month difference in the occurrence of the two peaks, with the peak PRD OCC_LT_ occurring in April and the peak NDVI occurring in July, suggesting that vegetation growth in this region is a weak driver of ozone. In the northern part of the YRD, as a major wheat-rice-growing area, the cultivation cycle of cash crops has a strong influence on NDVI [52], and the maturation of cash crops for harvesting causes the NDVI during May–June in YRD to undergo a period of rapid decline, and the rapid decrease in vegetation is accompanied by a significant decrease in OCC_LT_, so the relationship between changes in OCC_LT_ and vegetation growth is linear. At QTP, due to the low vegetation cover related to the climate, there is no significant relationship between OCC_LT_ and vegetation growth trends.

By comparing the correlation between NDVI and OCC_LT_ among different regions, as shown in Table 5, it can be seen from the fitted relationship equations in most regions that NDVI and ozone changes show a clear linear correlation. The consistency between OCC_LT_ and NDVI trends is highest in UCS, with an r as high as 0.87, which indicates that changes in OCC_LT_ in this region are significantly influenced by vegetation growth. Followed by MLS and BTH, the r exceeds 0.7, indicating that plant growth is an important influencing factor driving the increase in OCC_LT_ in this region. In SCB and YRD, the growth of vegetation cover also has some influence, while there is no significant correlation between the two in the PRD and QTP regions.

### 4.4. Temperature Effect

The results regarding the temperature of six layers in the troposphere from 1000 to 750 hPa were extracted using ERA-5 reanalysis data, and the temperature trends of different altitude layers in the lower troposphere during 2008–2020 are shown in Figure 12. During 2008–2020, the temperature of all six altitude layers showed an increasing trend and the change trend was basically the same, but the rate of increase varied among different altitude layers. The atmospheric temperature in the 1000 mb altitude layer increases faster with a fitting coefficient of 0.045, and the overall temperature increase in this altitude layer is 0.63 °C. With increasing altitude, the temperature growth rate from the 950 hPa altitude layer to the 750 hPa altitude layer gradually slows down, the fitting coefficient decreases from 0.038 to 0.029, and the temperature growth ranges from 0.41 to 0.53 °C. Figure 13 further shows the spatial variation in temperature from year-to-year in the 1000 hPa altitude layer, and it can be seen that the temperature of the whole region of China shows an increasing trend, which undoubtedly provides the conditions for the increase in OCC_LT_. Temperature, as the main driver of photochemical reactions in the process of ozone pollution generation, directly affects the photochemical reaction rate of precursors. Combined with the increase in HCHO, described in Section 4.2, it is clear that the increase in temperature causes the photochemical reaction rate of converting VOCs to ozone to increase, and thus more ozone can be produced in the same period of time. Therefore, the increase in the temperature of the lower troposphere is an important meteorological factor leading to the increase in OCC_LT_ in China.

### 4.5. PBLH Effects

In this study, the monthly and annual average values of PBLH over China were calculated using the ERA-5 meteorological reanalysis data. Then, the correlation between PBLH and OCC_LT_ was analysed using the averaged results. Figure 14 is a scatter plot of the monthly mean values of OCC_LT_ and PBLH in China from 2005 to 2020, and the r of the two is 0.902, which demonstrates a high positive correlation. Similar to other studies, the results illustrate the contribution of elevated boundary layer height to OCC_LT_. Figure 15 shows the annual average spatial distribution of PBLH from 2005 to 2020. The slope of the PBLH in the Chinese region ranges from −2 to 2. The PBLH in most regions show an increasing or slightly decreasing trend, and only some regions in QTP (Qinghai) exhibit a significant decrease. The regions with increasing PBLH also correspond to the regions with significantly increasing or higher OCC_LT_ shown in Figure 6a, i.e., the four regions of MSL, BTH, COC, and YRD. For the other regions, although there is a small decrease in the PBLH, the increase in temperature (Figure 13) still eventually leads to an increase in the OCC_LT_ that is smaller than that of the above four regions. Among the meteorological driving factors leading to OCC_LT_ growth, the effect of temperature is greater than that of PBLH. However, once the PBLH and temperature in a certain region are increased simultaneously, the growth of OCC_LT_ will be further enhanced.

## 5. Conclusions


The vertical distribution characteristics of tropospheric OCC in China from 2005 to 2020 were analysed using OMI ozone profile products. The results show that the OCC_LT_ shows an increasing trend from 2005 to 2020, with a mean year-on-year change of 0.143 DU, and a decreasing trend in the mid-troposphere, with a mean year-on-year change of −0.091 DU; OCC across the troposphere rose by 2.52 DU or 7.9% There is a significant negative correlation between the slopes of the OCCs at the two high layers (with an r of −0.891), which indicates that the OCC_LT_ is affected by the ozone transmission from the mid-troposphere.The variation in OCCs in the different altitudes of the troposphere is characterised by obvious seasonal changes, with the OCC_LT_ being higher in spring (8.22 ± 1.05 DU) and summer (8.80 ± 1.29 DU) than in autumn (6.38 ± 0.77 DU) and winter (5.70 ± 0.48 DU); the extreme values of the OCC_LT_ occur in May or June and peak in the middle troposphere in autumn (9.25 ± 1.19 DU) and winter (10.50 ± 0.45 DU). The mid-troposphere is strongly influenced by topographic conditions; in the upper troposphere it rises continuously in spring (up to 11.84 ± 1.30 DU) and decreases during the rest of the season, while the upper troposphere OCC shows a consistent latitude-dependent trend.Analysis based on multi-source data shows that the changes in OCC_LT_ are closely related to ozone precursors, vegetation cover, air temperature, and PBLH. The energy conservation and emission reduction policies in China in recent years have led to a large reduction in NO_x_, and the increase in VOCs due to natural factors has weakened the titration of ozone production and contributed to the increase in OCC_LT_. The NDVI shows a fluctuating upward trend, and the vegetation growth shows a positive correlation with the OCC_LT_, with an r > 0.6 in most regions of China; the temperature of the lower troposphere in China increases significantly (+0.63 °C), which strengthens the photochemical reaction of ozone. In addition, the increase in the PBLH also plays a positive role in the increase in the OCC_LT_.


In summary, the mechanism of influence of tropospheric ozone is complex, and is not the result of a single factor; both natural factors and climate change make important contributions to increases in tropospheric ozone and cannot be ignored. At the same time, oxygen pollution precursors should be controlled in a synergistic manner, and a single effort to control one of these factors may lead to an increase in ozone. This study recommends the incorporation of more long-term and diversified monitoring tools and the implementation of more comprehensive control measures, which are essential to control ozone pollution and to safeguard health and the balance of the ecosystem.

## Figures and Tables

**Figure 1 ijerph-19-12653-f001:**
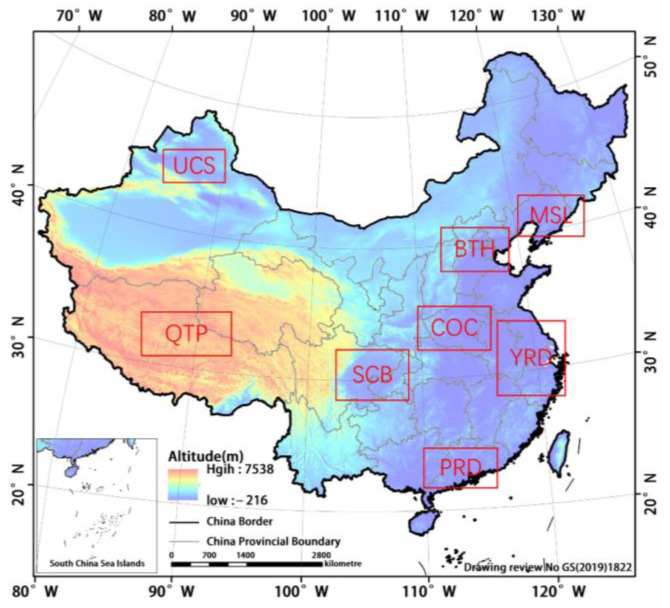
Elevation map of the study area.

**Figure 2 ijerph-19-12653-f002:**
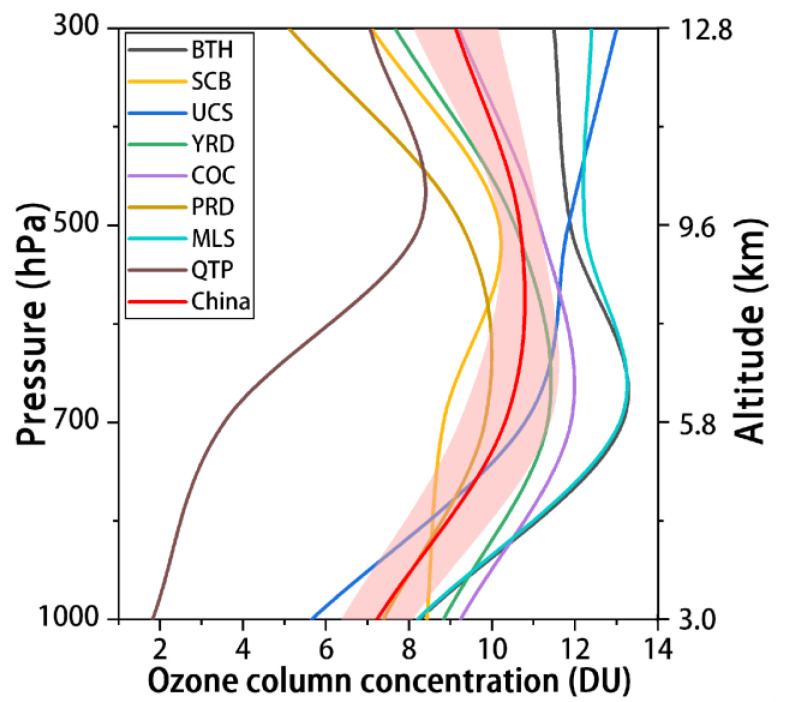
The average tropospheric ozone profile over the study area.

**Figure 3 ijerph-19-12653-f003:**
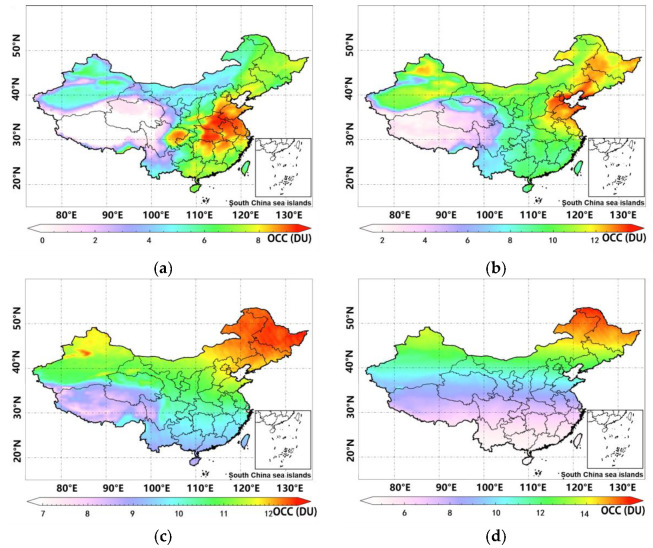
Distribution characteristics of tropospheric ozone at different heights: (**a**) characteristics of ozone distribution at 0–3 km in mainland China; (**b**) characteristics of ozone distribution at 3–5.8 km in mainland China; (**c**) characteristics of ozone distribution at 5.8–9.6 km in mainland China; and (**d**) characteristics of ozone distribution at 9.6–12.8 km in mainland China.

**Figure 4 ijerph-19-12653-f004:**
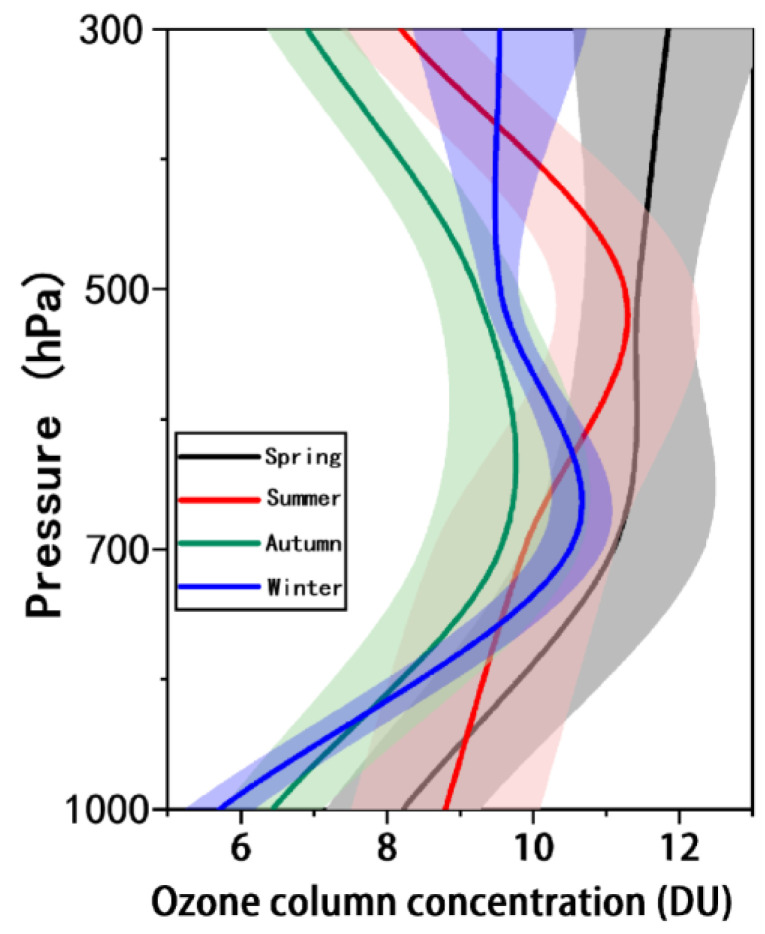
Seasonal distribution characteristics of tropospheric ozone profiles in mainland China (shaded areas indicate the standard deviation ranges; spring is March to May, summer is June to August, autumn is September to November, and winter is December to February).

**Figure 5 ijerph-19-12653-f005:**
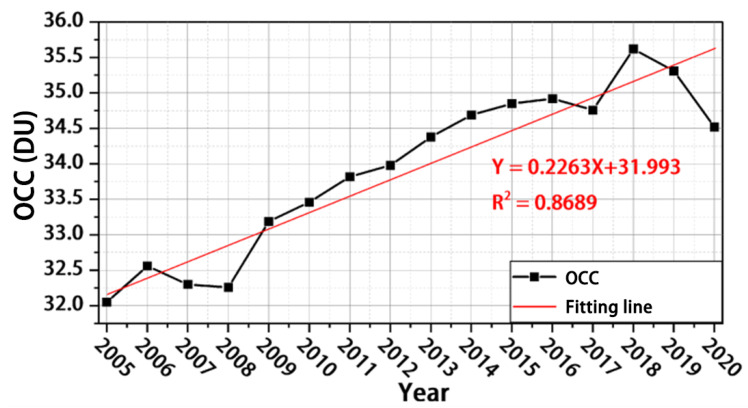
Interannual variation in tropospheric OCC in mainland China.

**Figure 6 ijerph-19-12653-f006:**
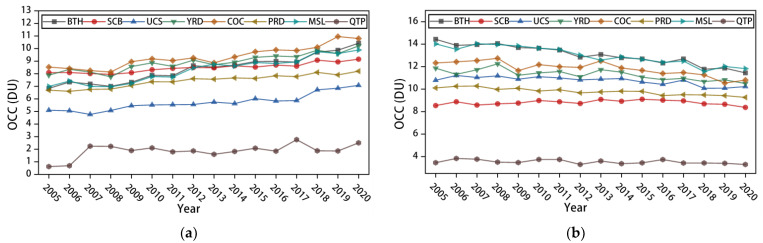
Interannual variation characteristics of OCC at different altitudes in each study area: (**a**) interannual variation characteristics of OCC at 0–3 km; (**b**) interannual variation characteristics of OCC at 3–5.8 km; (**c**) interannual variation characteristics of OCC at 5.8–9.6 km; (**d**) interannual variation characteristics of OCC at 9.6–12.8 km.

**Figure 7 ijerph-19-12653-f007:**
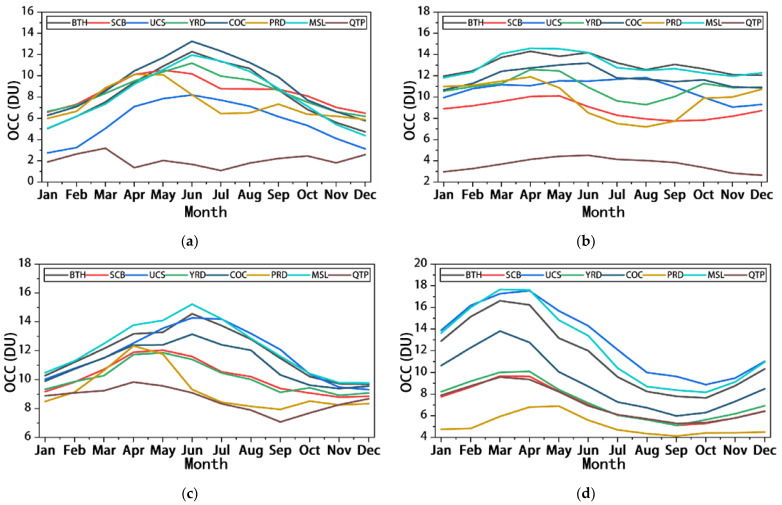
The monthly variation trend of OCC at different troposphere levels in China: (**a**) the monthly average variation trend of OCC at 0–3 km; (**b**) the monthly average variation trend of OCC at 3–5.8 km; (**c**) the monthly average variation trend of OCC at 5.8–9.6 km; and (**d**) the monthly average variation trend of OCC at 9.6–12.4 km.

**Figure 8 ijerph-19-12653-f008:**
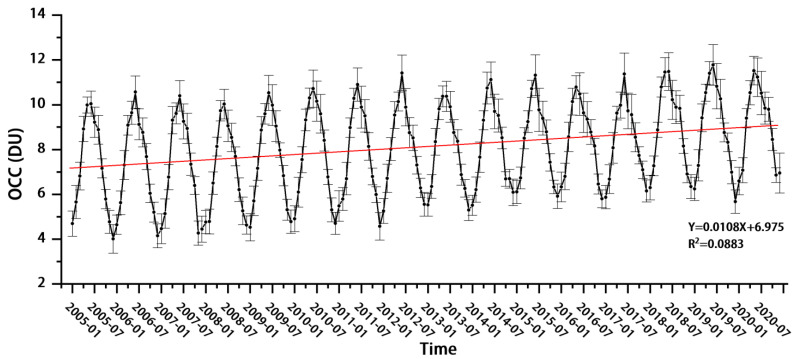
Variation trend of the annual mean value of the OCC_LT_ over the entire Chinese land area from 2005 to 2020.

**Figure 9 ijerph-19-12653-f009:**
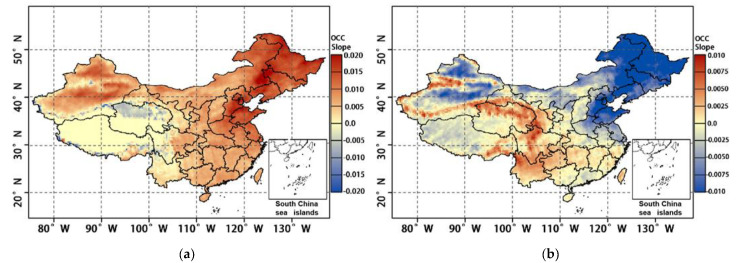
Slope results for OCC: (**a**) OCC slope of 0–3 km; (**b**) OCC slope of 3–5.8 km.

**Figure 10 ijerph-19-12653-f010:**
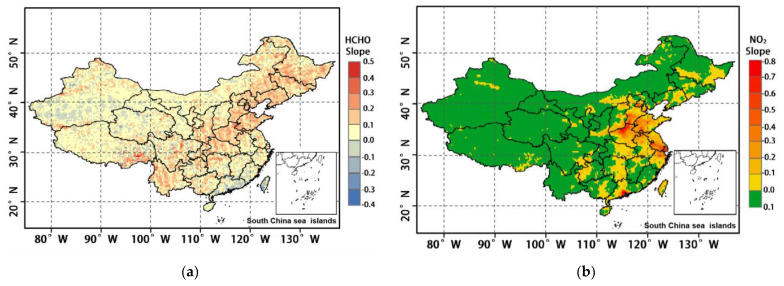
Slope variation trend of ozone precursors: (**a**) slope variation trend of HCHO; (**b**) slope variation trend of HCHO.

**Figure 11 ijerph-19-12653-f011:**
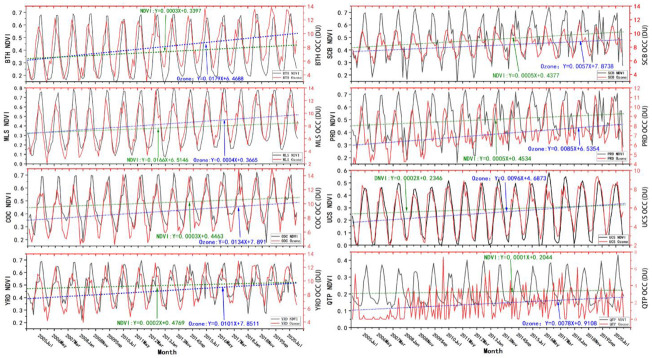
Interannual variation in NDVI and OCC_LT_ in mainland China.

**Figure 12 ijerph-19-12653-f012:**
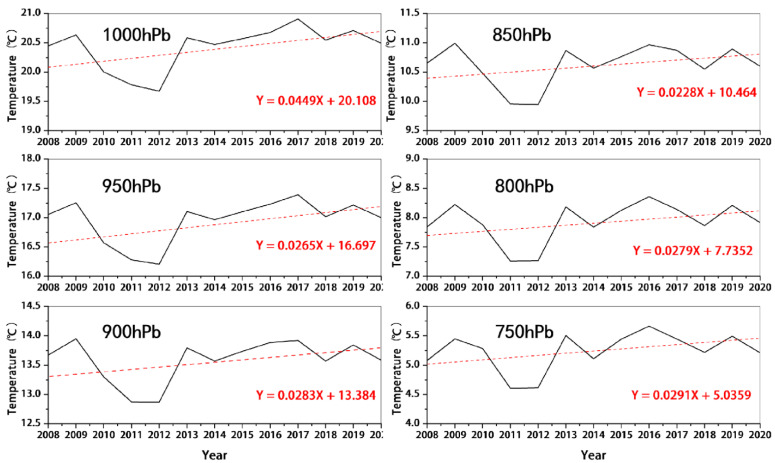
Variation trend of temperature at different altitudes in the lower troposphere from 2008 to 2020.

**Figure 13 ijerph-19-12653-f013:**
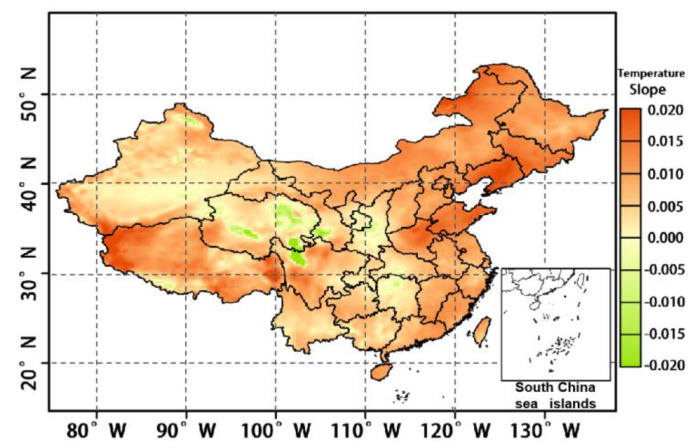
The 1000 hPa temperature distribution in the lower troposphere from 2008 to 2020.

**Figure 14 ijerph-19-12653-f014:**
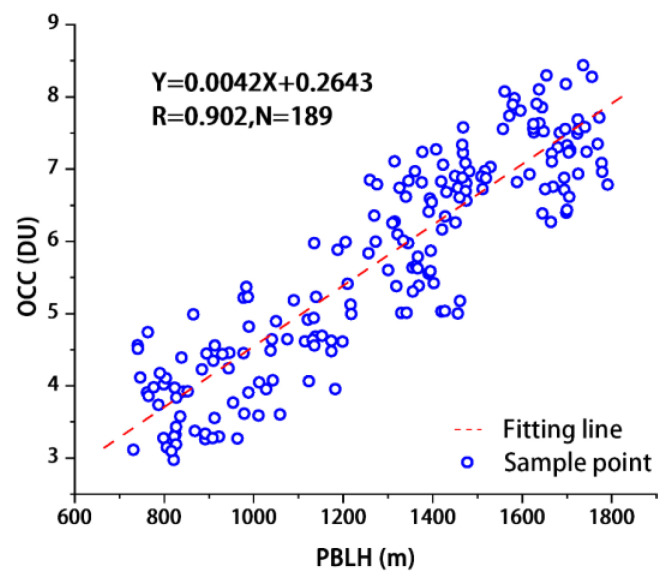
Scatter plot of monthly mean values of OCC_LT_ and PBLH in China from 2005 to 2020.

**Figure 15 ijerph-19-12653-f015:**
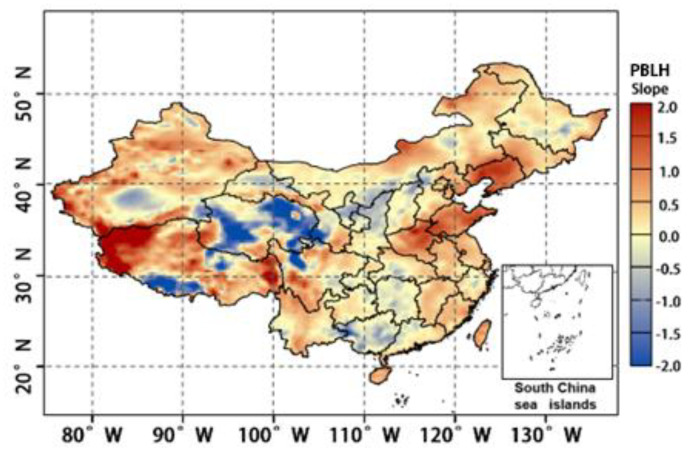
Variation map of the annual average spatial distribution of PBLH from 2005 to 2020.

**Table 1 ijerph-19-12653-t001:** Spatial ranges of the study area.

Urban Agglomerations	Spatial Range
Beijing-Tianjin-Hebei	114–119.5° E, 37–42° N
Yangtze River Delta	116.5–122° E, 27.5–34° N
Urumqi-Changji-Shihezi	86.5–89° E, 42.5–45° N
Qinghai-Tibet Plateau	73–105° E, 36–40° N
Pearl River Delta	111–115° E, 22–25° N
Sichuan Basin	103.5–109° E, 28.5–32° N
Centre of China	111–116° E, 32–37° N
Mid-Southern Liaoning	121–125° E, 38.5–42.5° N

**Table 2 ijerph-19-12653-t002:** Altitude layer of OMI ozone profile and the corresponding atmospheric pressure.

High Level	Altitude (km)	Atmospheric Pressure (hPa)
Layer-15	9.6–12.8	300
Layer-16	5.8–9.6	500
Layer-17	3–5.8	700
Layer-18	0–3	1000

**Table 3 ijerph-19-12653-t003:** MYC of OCCs at different tropospheric altitudes (DU).

Area	0–3 km	3–5.8 km	5.8–9.6 km	9.6–12.4 km
BTH	0.239	−0.200	−0.014	−0.034
SCB	0.074	−0.011	0.032	0.009
UCS	0.132	−0.063	−0.012	−0.022
YRD	0.155	−0.091	0.001	−0.008
COC	0.166	−0.113	−0.015	−0.013
PRD	0.105	−0.061	0.005	−0.002
MSL	0.205	−0.163	−0.007	−0.035
QTP	0.064	−0.019	0.053	−0.007
Mean	0.143	−0.091	0.005	−0.014

**Table 4 ijerph-19-12653-t004:** Correlation coefficient of the OCC slope at different altitudes and at 3–5.8 km.

Altitude Layer (km)	Number of Samples	Correlation Coefficient
53.3–57	10,766	0.264
48–53.3		0.334
42.8–48		0.497
40–42.8		0.457
36–40		0.319
33.7–36		0.374
31.4–33.7		0.494
26.7–31.4		0.429
24–26.7		−0.283
21–24		−0.261
18.8–24		−0.001
16.7–18.8		0.583
14.3–16.7		0.787
12.4–14.3		0.789
9.6–12.4		0.805
5.8–9.6		0.284
0–3	9290	−0.891

**Table 5 ijerph-19-12653-t005:** Correlation between OCC_LT_ and NDVI.

Area	r	Fitted Formula
UCS	0.875	y = 26.0044x + 3.05747
MSL	0.793	y = 16.115x + 4.23403
COC	0.785	y = 16.115x + 4.23403
BTH	0.746	y = 18.2164x + 4.18514
YRD	0.630	y = 16.9844x + 4.32518
SCB	0.613	y = 9.44341x + 5.86769
PRD	0.330	y = 4.39351x + 6.11428
QTP	−0.184	y = −6.23702x + 2.90587

## Data Availability

The data used to support the findings of this study are available from the corresponding author upon request.

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
