# Peer review of "Analysis of Vertical Distribution Changes and Influencing Factors of Tropospheric Ozone in China from 2005 to 2020 Based on Multi-Source Data"

_ijerph, 2022, doi:10.3390/ijerph191912653_

Round 1
Reviewer 1 Report
In this paper, the authors analyzed vertical distribution changes and influencing factors of tropospheric ozone in China from 2005 to 2020 based on the ozone profile product of the ozone monitoring instrument (OMI) on Arua, NASA’s Earth observation system satellite. During the study period, they extensively investigated the distribution characteristics of tropospheric ozone column concentration (OCC). The study gives us an insight into how the variation of OCC at altitudes is characterized by seasonal changes. The comprehensive work of the project is reflected in the thorough and well-written manuscript. The manuscript is clear and relevant to the field. The introduction, methodology, results, and discussion are adequately represented. The manuscript's results appear to be statistically significant and, given the resources, would be reproducible. The conclusions are consistent with the evidence seen in the analysis section of the paper.
The English language and the grammar presented in the paper are consistent and good. Some typing errors remain, but the changes required are all minor edits.
I have general and some specific comments. I would appreciate it if the authors addressed them.
Line 17-21: Sentence needs to be rephrased. If possible, avoid using lengthier sentences.
Line 26-32: The sentence needs to be rephrased. If possible, avoid using lengthier sentences.
Line 40: here by “troposphere” does the authors mean to say lower troposphere and PBL?
Even though the authors mentioned OCC in the abstract, it is better that OCC is defined and stated one more time in the introduction.
The authors used a mixture of regular and subscripts throughout the text for NO(x) and O(3). I highly recommend maintaining uniformity throughout the text. Using subscripts for NO(x) and O(3) is standard practice.
Line 62: is it just 8-h or 9-h average daily concentration?
Line 91: double (,,)
Line 94-97: Even though there is a brief description of the planetary boundary layer (PBL) in the introduction, there is no adequate explanation of how PBL can affect ozone levels or the seasonal pattern of the PBL and its influence on ozone concentrations.
Line 133-143: Entire subsection 2.1 needs to be rephrased. Avoid using lengthier sentences.
Line 147: If possible, kindly provide a weblink to the data source.
Line 163: Is it 2004 or 2005?
Kindly specify what DU stands for the readers.
Line 174: maintain uniformity (space or no space) “_x_”
Line 215: slope
Line 246: Figure 3(a)”space”shows
Line 263-266: This line needs further elaboration
Line 268-271: This line needs further clarification
Line 275: vertical space before the figure
Line 281-290: subsection 3.1.2 needs further elaboration
Line 322: vertical space before the figure
Line 363: Sentence seems incomplete.
Line 376, 378, 387, 391: period after the citation/reference number
Line 399: Figure 8, if possible, make the data line slightly bolder
Line 509: Figure 11, if possible, increase the figure dimensions. It would be hard for the readers to read the data trend.
Line 560-562: This sentence needs to be further explained.
Line 562: If possible, use dark color points. Blue dots are barely visible.
Line 568-569: This line needs some elaboration.
Line 571: OCC
Given the number of acronyms used in the manuscript, I strongly advise the authors to include a list of abbreviations at the end of the paper (before references).
More references, especially for PBL -Ozone interaction, can be added to enrich the literature and consolidate the claims.
e.g. (Karle et al. 2020), (Hallar et al. 2021) and (Karle et al. 2021)
Karle, N.N.; Mahmud, S.; Sakai, R.K.; Fitzgerald, R.M.; Morris, V.R.; Stockwell, W.R. Investigation of the Successive Ozone Episodes in the El Paso–Juarez Region in the Summer of 2017. Atmosphere 2020, 11, 532.
Hallar, A. Gannet et al. 2021. “Coupled Air Quality and Boundary-Layer Meteorology in Western U.S. Basins during Winter: Design and Rationale for a Comprehensive Study.” Bulletin of the American Meteorological Society 102(10): E2012–33.
Karle, Nakul N. et al. 2021. “Multi-Scale Atmospheric Emissions, Circulation and Meteorological Drivers of Ozone Episodes in El Paso-Juarez Airshed.” Atmosphere 12(12): 1575.
Reviewer 2 Report
This paper presents an analysis of vertical distribution of tropospheric ozone in China from 2005-2020 based on OMI data. Additionally, the analysis of factors influencing the changes in ozone content in the lower troposphere was performed. This an important issue especially in view of the increasing levels of ground-level ozone. Nevertheless, I have a few comments regarding this article:
In general, the whole article was written carelessly in terms of language, which makes it difficult for reading. The lack of comma, the lack of full stop, repetitions, use of lowercase letters instead of capital letters, using a semicolon instead of a full stop, too long sentences, incorrectly used abbreviations are the most common mistakes that I noticed. I recommend Authors to read the article carefully again and correct it taking into account the above remarks. In addition, I recommend proofreading in terms of English language.
Detailed comments:
Abstract:
Please adjust the size of the abstract to the MDPI guidelines – max 200 words.
Line 26: Please explain the abbreviations used for the first time in the text (NOx, VOC).
Line 31: Please delete the repetition.
Introduction:
Sometimes O3, sometimes ozone is used. Please use an abbreviation when „ozone” is used for the first time and then use it consistently throughout the text. Additionally please use subscripts Whiting O3.
Line 57: Please don’t use full stop before square bracket with references. Note that such a situation occurs several times in the text
Line 81: Please expand the TOC abbreviation.
Line 84: Authors used full name of VOC. It is without a sense.
Line 94: Generally titration effect is caused by NO. The removal of O3 through reaction with NO is presented by the re action: NO + O3 → NO2 + O2
Line 101: Please use the OCC abbreviation.
Line 126-131: Please avoid too long sentences.
In the subsections where an averages values are calculated (e.g. seasonal, monthly), please add the standard deviation values.
Line 218: Please explain which elements?
Line 236: Please write so that it Gould be understandable what is going on. Which concentrations? Probably not BTH and MLS but O3.
Line 232-241: In the description of Figure 2 the Authors use the [km] instead of [hPa] to present value of altitude. Please add additional axis with [km] unit.
Line 243: Please add an appropriate description of the Figure 2.
Line 253: What do you mean: „average poster height”?
Line 286, 287, 289: Please correct „winter” word.
Figure 4. Please correct description of x axis
Line 320-322: Please ex pand description of „ cyclical fluctuations caused by climatic factors”.
Line 363: Why you use „The trend” in the middle of the sentence?
Line 369-374: Please correct subscripts.
Line 390: Why temperature increase the natural emission prevention of VOC? Why prevention?
Line 403: Please be consistent in using abbreviations (OCC).
Line 404: please explain the potential causes of difference of QTP from the rest of the places.
Line 439-445: Please correct this sentence.
Line 471: Please correct the factor of O3 titration effect.
Line 479: Please explain NDVI abbreviation.
Line 483-488: Please correct this sentence.
Line 511-518: This sentence is too long, please correct it.
Line 539: Please take care to use capital letters after the full stop.
Line 544-548: Please correct this sentence.
Conclusions:
Line 570-577, 583: Please use capital letters using OCC abbreviation.
The Conclusions are rather weak. Please improve it adding more detailed results of your work.
